# Disentangling Controlled Effects for Hierarchical Reinforcement Learning

**Oriol Corcoll**  ORIOL.CORCOLL.ANDREU@UT.EE  and  **Raul Vicente**  RAUL.VICENTE.ZAFRA@UT.EE
*Institute of Computer Science*
*University of Tartu*

**Editors:** Bernhard Schölkopf, Caroline Uhler and Kun Zhang

## Abstract

Exploration and credit assignment are still challenging problems for RL agents under sparse rewards. We argue that these challenges arise partly due to the intrinsic rigidity of operating at the level of actions. Actions can precisely define how to perform an activity but are ill-suited to describe what activity to perform. Instead, controlled effects describe transformations in the environment caused by the agent. These transformations are inherently composable and temporally abstract, making them ideal for descriptive tasks. This work introduces CEHRL[1], a hierarchical method leveraging the compositional nature of controlled effects to expedite the learning of task-specific behavior and aid exploration. Borrowing counterfactual and normality measures from causal literature, CEHRL learns an implicit hierarchy of transformations an agent can perform on the environment. This hierarchy allows a high-level policy to set temporally abstract goals and, by doing so, long-horizon credit assignment. Experimental results show that using effects instead of actions provides a more efficient exploration mechanism. Moreover, by leveraging prior knowledge in the hierarchy, CEHRL assigns credit to few effects instead of many actions and consequently learns tasks more rapidly.

**Keywords:** unsupervised reinforcement learning, reinforcement learning, causality

## 1. Introduction

Value-based methods for reinforcement learning (RL) (Sutton and Barto, 1998) have achieved impressive results in environments with dense rewards (Mnih et al., 2013). These methods learn by estimating the causal effect of actions on the reward function. However, this type of learning is particularly ineffective in environments with sparse rewards where no rewards are given for long periods of time, thus requiring to collect vast amounts of experience, each providing little to no learning. A promising solution is to use hierarchical RL (Sutton et al., 1999) to learn reusable skills. Skill discovery research has focused on methods based on information theory (Florensa et al., 2017; Eysenbach et al., 2018; Sharma et al., 2019) or intrinsic motivators (Kulkarni et al., 2016; Nachum et al., 2018).

In contrast, we are motivated by studies in the field of developmental psychology, indicating that children use the consequences of their actions to learn how to control their environment (Goodman et al., 2007; Buchsbaum et al., 2012, 2015). We conceptualize skills as useful changes in the environment; thus, we propose agents that learn by estimating the causal effect of an action on the state. Intuitively, these effects describe ways in which the agent can change its environment. Since the agent does not control every change, we adopt counterfactual measures from causal literature (Pearl, 2009; Halpern, 2016) to disentangle effects caused by the agent from effects caused by other dynamics, e.g., other agents, wind, etc.

---

1. pronounced 'ciril'.

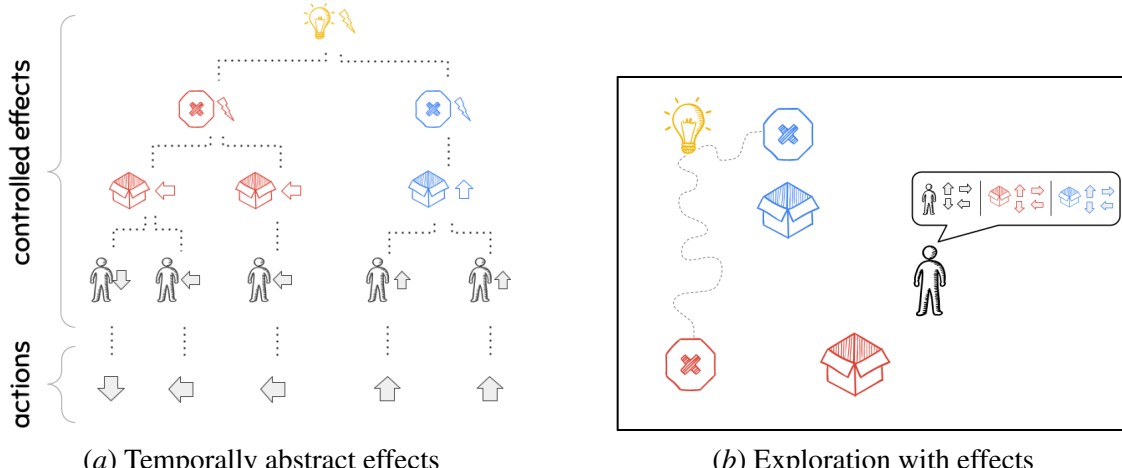

(*a*) Temporally abstract effects  (*b*) Exploration with effects

Figure 1:  a) A complex effect like turning on a light requires simpler effects like activating a switch, all the way down to actions; creating an implicit hierarchy of effects. b) To get a reward, the agent needs to turn on a light by moving boxes to their switches. Instead of performing random actions until the light turns on, CEHRL's exploration is based on performing random controlled effects (e.g. move box) to discover novel and more complex effects.

A key aspect of controlled effects is their compositionality, i.e., to perform a complex effect, an agent needs to combine simpler effects, which are also composed of more simple effects, all the way down to actions. Figure 1(*a*) illustrates the compositional nature of controlled effects, making them temporally abstract. In other words, effects may take several, possibly variable, number of actions to be performed (Sutton et al., 1999).

Temporal abstraction promises to ease the credit assignment problem by decoupling the agent from a fine-grained time scale. To this end, we introduce CEHRL (Controlled Effects for Hierarchical RL), a method that builds an implicit hierarchy of controlled effects serving a twofold purpose. First, instead of exploring the environment via random actions, CEHRL relies on random effect exploration. Consider the scenario presented in Figure 1(*b*), where an agent has already learned some basic effects like moving the agent and the boxes. Turning on the light is more likely to happen the more boxes are moved. Consequently, random effect exploration is based on the idea that the agent can discover and learn more complex effects by combining basic effects, continuously enriching the hierarchy in an unsupervised manner. Second, a rich hierarchy facilitates the learning of task-specific behavior efficiently. By reusing prior knowledge, the agent quickly discovers and learns from reward.

CEHRL relies on counterfactuals to identify the causal effect of an agent on the environment. Then, it uses a Variational Autoencoder (VAE) (Kingma and Welling, 2013) to represent the hierarchy of transformations an agent can cause on the environment. A goal-conditioned policy translates controlled effects from the VAE into a sequence of actions. The VAE and policy work in conjunction to explore the environment by 1) randomly selecting effects from the hierarchy as goals for the policy and 2) using newly discovered controlled effects as learning signal to the VAE. Finally, to learn a particular task, CEHRL trains a state-effect value function to score how valuable a given controlled effect is to maximize the environment's reward.

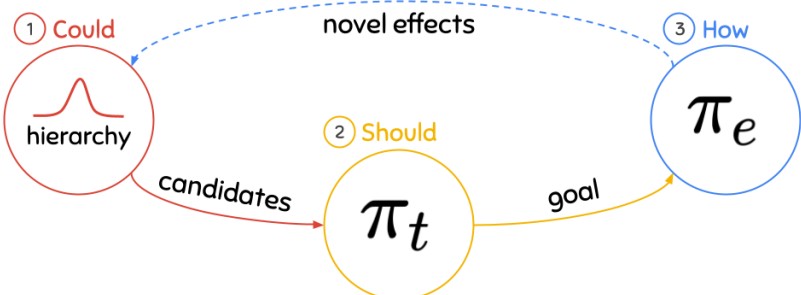

Figure 2: CEHRL has three components: a generative model of what **could** be done to the environment; a task policy that decides which effect **should** be performed next; and an effect-conditioned policy that decides **how** to perform an effect. Note that different choices of $\pi_t$ can promote exploratory or task-specific behavior.

Our main contributions are:
- We establish a link between the advantage function and causality.
- A novel method based on counterfactuals to identify effects controlled by the agent.
- An unsupervised approach to continually learn a hierarchy of controlled effects.
- An algorithm that uses this hierarchy to rapidly learn task-specific behavior from rewards.

## 2. Disentangling Controlled Effects

Pearl et al. (2016) provide an intuitive definition of cause-effect relations: "*A variable X is a cause of a variable Y if Y, in any way, relies on X for its value*". For example, if the life of an agent relies on eating food, eating food has a causal effect on the agent's life. Actual causality proposed in Halpern (2016) studies causal relations between individual events of $X$ and $Y$. For example, would taking action $a$ have an effect on the agent's life at a particular state? Figure 3 shows the causal graph of a typical RL setting, where a state $s$ has a causal effect on both the agent's choice of action and the next state $s'$. Similarly, an action has an effect on the next state. In our experiments, we assume deterministic tran-

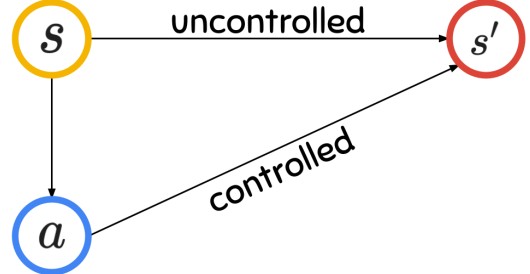

Figure 3: Controlled and uncontrolled effects.

sitions, discrete actions and $s \in \mathbb{N}^p$ is a $p$-dimensional vector. We define the total effect $e_t$ as the change in the environment's state when transitioning from $s$ to $s'$ due to taking action $a$ i.e. $e_t(s, a) \equiv s' - s$. This section aims to disentangle controlled effects $e_c$ caused by the agent's action from total effects.

The individual causal effect (ICE) of a variable $X$ on another variable $Y$ can be measured by comparing counterfactual worlds (Pearl, 2009)

$$ICE^x_{Y_i} \equiv Y_i^x \neq Y_i^{\tilde{x}} \,, \tag{1}$$

where $Y_i^x$ reads as "what would the value of $Y_i$ be if $X$ had taken value $x$". Similarly, $Y_i^{\tilde{x}}$ describes the value of $Y_i$ when $X$ does not take the value of $x$. Intuitively, Eq. 1 compares the world where the event $x$ happened against an alternative world where event $x$ had not happened. The *fundamental problem of causal inference* (Holland, 1986) states that this last world cannot be observed, therefore needs to be imagined. Halpern and Hitchcock (2011, 2014) propose to compare what happened with what normally would happen by constructing a normative world

$$ICE_{Y_i}^x \approx Y_i^x - \beta_{Y_i} \, , \tag{2}$$

where $\beta_{Y_i}$ is the value $Y_i$ would normally take. Of course, such value is contingent to the notion of normality used, which is for us to define. Note that contrary to Eq. 1, this formulation uses the magnitude and direction of the effect, thus computing the difference between worlds.

A common use in reinforcement learning of this formulation is to compute the causal effect of an action on the return $G$

$$
\begin{aligned}
G^a - G^{\tilde{a}} &= Q(s,a) - V(s) \\
&= A(s,a).
\end{aligned}
\tag{3}
$$

$G_a$ is the return the agent would get if action $a$ were to be taken. Usually, $G_a$ is estimated using a state-action value function $Q(s,a)$ and the choice of normality for $G_{\tilde{a}}$ is the expected return estimated with the state-value function $V(s)$. This choice gives the advantage function $A(s,a)$ which estimates the causal effect of an action on the return.

As described in Richard S. and Barto (2018), General Value Functions include general knowledge of the world, leaving the return as special case. Here, we follow the same idea and use Eq. 2 to identify the causal effect of an action on the state

$$e_c(s,a) \equiv e_t\left(s,a\right) - e_t\left(s,\tilde{a}\right), \tag{4}$$

where as before, $e_t(s,\tilde{a})$ needs to be imagined. Note that defining $\tilde{a}$ as a special *do-nothing* action would not work since an agent can still have an effect on the environment by doing nothing. For example, an agent doing nothing when an enemy is moving towards it would have a causal effect on its life. Our choice of normality is to compute $e_t(s,\tilde{a})$ as the most common world among every possible world, i.e. we compute $e_t(s,\tilde{a})$ as the mode over all actions and consequently $e_c(s,a)$ as

$$e_c(s,a) \equiv e_t\left(s,a\right) - \operatorname*{mode}_{a_i \in A}\left(e_t\left(s,a_i\right)\right). \tag{5}$$

In practice we do not have access to every world and cannot compute Eq. 5 directly. We approximate $e_t(s,a)$ using a neural network $\hat{e}_t(s,a)$ trained using experiences $(s,a,s')$ and loss

$$\mathcal{L}_{\hat{e}_t} = \text{MSE}\left(\hat{e}_t\left(s,a\right), e_t\right). \tag{6}$$

We discretize total effects to its nearest integer when computing Eq. 5 but train $\hat{e}_t$ with its continuous output. We provide more details of the training in section 3.

## 3. CEHRL

This section explores how controlled effects identified as in Eq. 5 can be used for temporal abstraction. Knowing the changes an agent can perform to the environment is valuable information invariant

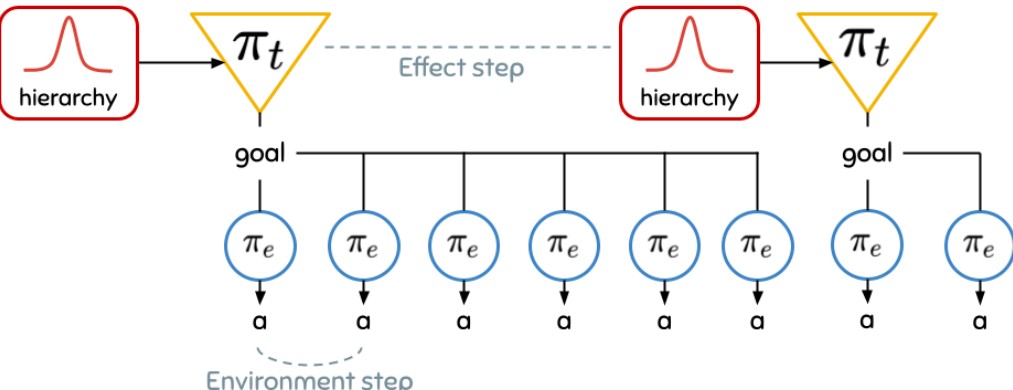

Figure 4: CEHRL's components as they are used for either unsupervised exploration and learning from reward. The use of the model will depend on how $\pi_t$ is implemented. The two policies of CEHRL work at different time scales: effect steps and environment steps.

to the environment's time scale. When operating at the level of actions, reward needs to propagate through many time steps. In the example from Fig. 1(a), the reward gotten when the light turns on would need to travel through the whole bottom of the hierarchy, i.e., actions. Instead, we aim to propagate reward through higher levels of the hierarchy, e.g., turn red and blue switches on. By doing so, we would be able to reuse prior knowledge and learn from few examples.

The following describes CEHRL (see Figure 2), an agent that uses controlled effects to learn temporally abstract effects to operate on an extended time horizon. CEHRL is a hierarchical method composed of a generative model $E$ to provide the different ways an agent could change the environment; a task policy $\pi_t(e_c^{\text{goal}}|s, \boldsymbol{e}_c)$ that decides what effect to perform next; and an effect-conditioned policy $\pi_e(a|s, e_c^{\text{goal}})$ that learns how to translate an effect into actions. CEHRL is trained in two stages, 1) the generative model and effect-conditioned policy are trained by exploring the environment to enrich the hierarchy; 2) the task-policy is trained to maximize the environment's reward by reusing prior knowledge from the hierarchy.

### 3.1. Controlled Effects for Unsupervised Exploration

CEHRL uses random effect exploration to continuously discover novel controlled effects. Random effect exploration is based on the idea that by combining controlled effects, an agent can discover novel and more complex effects. In other words, learned effects are stepping stones that facilitate the discovery of novel effects. Following Fig. 1(b), the agent would explore more efficiently by reusing previously learned effects which would ease the learning of more complex effects like turning a switch on. Random effect exploration samples effects from the learned distribution and sets these effects as goals for the effect-conditioned policy. Furthermore, the resulting controlled effects are used to train this distribution further, creating a continuous learning cycle.

**Acting:** to perform random effect exploration, we first sample $N$ candidate effects $\boldsymbol{e}_c = (e_c^1, \ldots, e_c^N)$ from the distribution of effects $E$; then the task policy selects one of these effects as goal $e_c^{\text{goal}}$; and

finally, we unroll the effect-conditioned policy until either the controlled effect for the current step matches the goal or more than $K$ actions have been performed. We store experiences of the form $(s, e_c^{\text{goal}}, a, s', e_c(s, a), r', d')$ in a prioritized replay memory $\mathcal{D}$ (Schaul et al., 2016). Instead of using the environment's terminal state $d$, we terminate the episode if either the environment's episode ends, the number of steps trying the goal exceeds a limit $K$ or the goal is reached. We ignore the extrinsic reward provided by the environment and give a reward of $1$ if the goal is reached or a punishment $P$ otherwise. Experiences are augmented with Hindsight Experience Replay (Andrychowicz et al., 2017). Additionally, we deal with the intrinsic imbalance between basic and complex effects by using a function $g(e_c)$ to compute the rarity of an effect and set it as initial priority to each experience.

**Learning:** here we describe CEHRL's components, the data they require and their training.
- *Total effects model:* to be able to compute $e_c(s, a)$ from Eq. 5, we train the network $\hat{e}_t(s, a)$ with the loss defined in Eq. 6 using samples $(s, a, s')$ from the replay memory.

- *Distribution of effects:* we train a Variational Autoencoder (Kingma and Welling, 2013) to approximate the distribution of controlled effects present in the replay memory. This model is not conditioned on state nor actions, making effects reachable from any state.

- *Effect-conditioned policy:* we train a neural network $Q_e(s, e_c^{\text{goal}}, a)$ to learn the state-effect-action value function using DQN (Mnih et al., 2013) with prioritized experience replay (Schaul et al., 2015), Double Q-learning (van Hasselt et al., 2015) and Dueling Networks (Wang et al., 2015); and use this network as effect-conditioned policy

$$\pi_e(a|s, e_c^{\text{goal}}) = \arg\max_{a \in A} Q_e(s, e_c^{\text{goal}}, a). \tag{7}$$

- *Task policy:* to bias exploration as little as possible, our choice of task policy is to choose effects following a uniform distribution

$$\pi_t(e_c^{\text{goal}}|s, \boldsymbol{e}_c) = \text{uniform}(\boldsymbol{e}_c). \tag{8}$$

We deem a goal achieved using the function has_achieved($e_c^{\text{step}}, e_c^{\text{goal}}$) $= |e_c^{\text{step}} - e_c^{\text{goal}}| < T$, where $T$ is a threshold fixed before training. In our experiments, we start by training the total effects model only, then we incorporate the distribution of effects to the training, and finally the effect-conditioned policy. See algorithm 1 and the supplementary materials for more details on the training, hyperparameters and network architectures used in our experiments.

### 3.2. Controlled Effects for Learning from Reward

In the previous section we chose to implement the task policy as a uniform sampling over effects so our exploration is as unbiased as possible. Here, we want to learn how to bias the agent so as to maximize a reward function. We do this by learning a Q-value function that estimates the value of state-effect pairs, thus the task policy is implemented as

$$\pi_t(e_c^{\text{goal}}|s, \boldsymbol{e}_c) = \arg\max_{e_c \in \boldsymbol{e}_c} Q_t(s, e_c). \tag{9}$$

We train $Q_t$ using DQN and a prioritized replay buffer with experiences $(s, e_c^{\text{goal}}, s', \boldsymbol{e'}_c, r', d)$. Here, $d$ is the terminal state provided by the environment and $\boldsymbol{e'}_c$ are the candidate effects produced by the

effect distribution for the next state. The effect-conditioned policy and task policy work at different time scales, i.e. an "effect step" is composed of multiple "environment steps", therefore states $s$ and $s'$ are consecutive effect steps but may not be consecutive environment steps. Since rewards are provided at every environment step, we accumulate rewards given between effect steps into $r'$. In this phase, we use pre-trained frozen models $\hat{e}_t$, $E$ and $\pi_e$ learned in the exploration phase. See algorithm 2 and supplementary materials for more details.

---

**Algorithm 1** Exploration phase

**Require:** $K; \mathcal{D}$
  Initialize $\hat{e}_t$; $E$; $\pi_e$; $s = s_0$; $d' = 1$
  **while** keep training **do**
    $\boldsymbol{e}_c \sim E$
    $e_c^{\text{goal}} = \pi_t(s, \boldsymbol{e}_c)$ # $\pi_t$ samples candidates uniformly

    **while** not $d'$ **do**
      $a \sim \pi_e(s, e_c^{\text{goal}})$
      $s', d = \text{step\_env}(a)$
      $e_c^{\text{step}} = e_c(s, a)$
      $h = \text{has\_achieved}(e_c^{\text{step}}, e_c^{\text{goal}})$
      $r' = \text{compute\_reward}(h)$
      $d' = (d \text{ or } (t > K) \text{ or } h)$
      $\text{add\_with\_her}(\mathcal{D}, g(e_c^{\text{step}}), (s, e_c^{\text{goal}}, a, e_c^{\text{step}}, s', r', d'))$
      $s = s'$

      $\text{train\_if\_needed}(\hat{e}_t, \mathcal{D})$
      $\text{train\_if\_needed}(E, \mathcal{D})$
      $\text{train\_if\_needed}(\pi_e, \mathcal{D})$
    **end while**
  **end while**

---

**Algorithm 2** Task-specific learning

**Require:** $K; \mathcal{D}$
  Load pre-trained $\hat{e}_t$; $E$ and $\pi_e$
  Initialize $\pi_t$; $s = s_0$; $d' = 0$
  **while** keep training **do**
    $\boldsymbol{e}_c \sim E$
    $e_c^{\text{goal}} = \pi_t(s, \boldsymbol{e}_c)$
    $s_g = s$

    **while** not $d'$ **do**
      $a = \pi_e(s, e_c^{\text{goal}})$
      $s', r, d = \text{step}(a)$
      $e_c^{\text{step}} = e_c(s, a)$
      $h = \text{has\_achieved}(e_c^{\text{step}}, e_c^{\text{goal}})$
      $r' = r' + r$
      $d' = (d \text{ or } (t > K) \text{ or } h)$
      $s = s'$
    **end while**

    **if** $h$ **then**
      $\text{add}(\mathcal{D}, (s_g, e_c^{\text{goal}}, e_c^{\text{step}}, s, \boldsymbol{e}_c, r', d))$
    **end if**
    $r' = r' * (1 - d)$

    $\text{train\_if\_needed}(\pi_t, \mathcal{D})$
  **end while**

---

## 4. Experiments

The following experiments provide empirical answers to the following questions: **a) credit assignment** - can controlled effects ease the task of credit assignment? **b) exploration** - How does exploration with random effects differ from using random actions? **c) cost** - how expensive is to operate with effects? **d) disentanglement** - Is there any benefit to use controlled effects? First, we show how using a hierarchy of controlled effects can help with the credit assignment and exploration problems. Secondly, we show how the cost of building this hierarchy is amortized when the number of tasks or their complexity increases.

We use a Grid World environment that includes multiple objects with different properties and ways of interaction. These objects are: a ball that can be picked and dropped, a chest where the agent can store the ball and a special target location where the agent can step into.

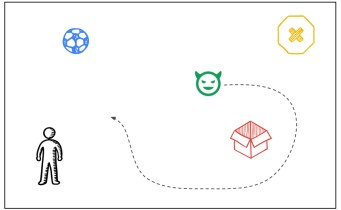

Additionally, we introduce a demon with its own dynamics to evaluate if CEHRL can disentangle controlled effects. We provide three tasks where reaching the special target becomes increasingly difficult:

- *T:* go to the target location.

- *BT:* go to the target location while carrying a ball

- *CBT:* pick ball, put it in the chest, and go to the target.

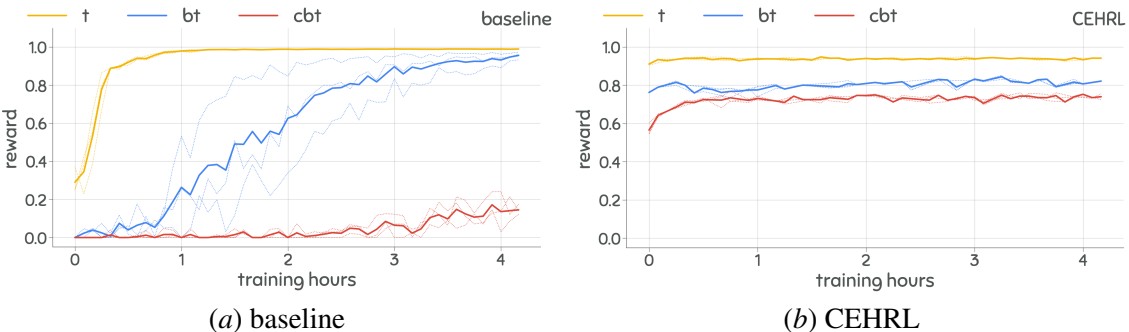

(a) baseline
(b) CEHRL

Figure 5: a) Reward achieved by the baseline in the three variants of our environment. b) Reward achieved by CEHRL. By using already learned effects, CEHRL can scale better with the task complexity. Bold lines denote mean reward over three seeds (dotted lines).

To implement this environment we use MiniGrid 2D (Chevalier-Boisvert et al., 2018). Although these kind of environments may seem simple, they are known to be hard to explore due to the combinatorial explosion of possible effects. In other words, it is not enough to move forward to explore new regions, the agent needs to learn to combine effects. Moreover, to separate the problems of representation learning and exploration, agents are provided with an entity-centric representation of the world as in Baker et al. (2019); Campero et al. (2021). A reward of one is given discounted by the time it took to solve the task, see supplementary material for more details. Although we list the set of possible effects in our results, they are not given to the agent; the agent has to learn them. We evaluate how CEHRL scales with the complexity of the task by comparing it to a DQN baseline. Note that the duration of a training step is different for each of the components in CEHRL and the baseline. To be fair to the baseline, we report training time.

### 4.1. Can controlled effects ease the task of credit assignment?

This experiment evaluates if CEHRL can ease the credit assignment problem by using a few effects instead of numerous actions. We train the task policy on the three variants of the environment using the pre-trained models $\hat{e}_t$, $E$ and $\pi_e$ from experiment 4.3.

Figure 5(a) shows the baseline achieving high reward in the first two tasks but cannot complete the more complex task. Moreover, it requires increasingly more time to complete each task, not scaling well with the task's complexity. On the other hand, CEHRL achieves high reward in the three tasks and needs 30 minutes to complete each task, scaling better with the task's complexity (Fig. 5(b)). We conjecture that learning the value of individual actions at every state requires a large number of samples, where using effects as abstractions reduces the space of values to model. Note that since we do not fine-tune $\pi_e$ further, CEHRL does not achieve an optimal reward.

These results suggest that using a hierarchy of effects simplifies the credit assignment problem and enables solving long-horizon tasks more efficiently. This is possible due to reusing prior knowledge acquired during exploration. Experiment 4.3 analyses the cost of learning these effects.

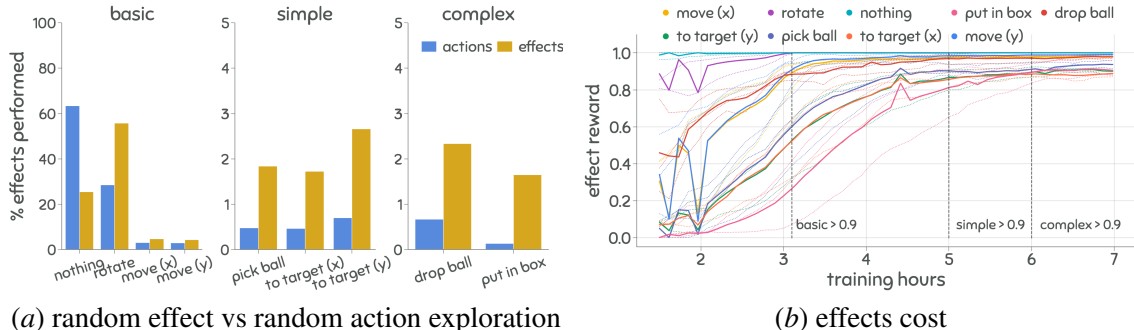

($a$) random effect vs random action exploration      ($b$) effects cost

Figure 6: a) Comparison between random action and random effect exploration. Note that basic effects are needed to perform simple and complex effects. b) Time taken to learn each effect. Each vertical line indicates when a group of effects reached a reward of $0.9$.

### 4.2. How does random effect exploration differ from using random actions?

To analyze random effect exploration, we record the effects performed by each method during 500K steps. We do not perform any training and use pre-trained models as in the previous experiment. Note that some effects are more difficult to perform than others, e.g. putting the ball in the chest vs moving forward. We use the following categorization: **basic effects** require usually one or two actions to be performed (these are usually effects on the agent), **simple effects** require more than one basic effect and **complex effects** require multiple simple effects. This choice is arbitrary but helps us better understand what each method is exploring.

An unbiased exploration method would perform effects as uniformly as possible. Note that a perfectly uniform exploration is not possible due to the compositional nature of effects i.e. to perform a complex effect the agent needs to perform multiple simple or basic effects. The results in Figure 6($a$) show that random effect exploration performs simple and complex effects almost three times more often than with random actions. The effect "nothing" is not needed to perform other effects; therefore, it happens less often. In contrast, other basic effects are central to simpler effects and consequently happening more often.

### 4.3. How expensive is to operate with effects?

We have shown that exploration and credit assignment benefit from operating with effects. Here, we study the cost of creating the hierarchy of effects using the method described in section 3.1. For this, we train models $\hat{e}_t$, $E$ and $\pi_e$ and record the reward achieved by each individual effect.

Figure 6($b$) shows that CEHRL can perform every basic effect (reward higher than $0.9$) after three hours of training and that simple and complex effects take five and six hours respectively. Comparing these results with experiment 4.1, we can see that CEHRL learns to perform every effect in six hours but takes five hours for the baseline to learn tasks $T$ and $BT$, and cannot learn task $CBT$. These results indicate that the cost of learning a hierarchy of effects is quickly amortized when the number of tasks or their complexity increases. Moreover, using effects enables the learning of tasks that the baseline struggles to learn. We conjecture that this is due to the implicit curriculum of learning causal effects, i.e. the learning of simpler effects makes easier to learn complex effects.

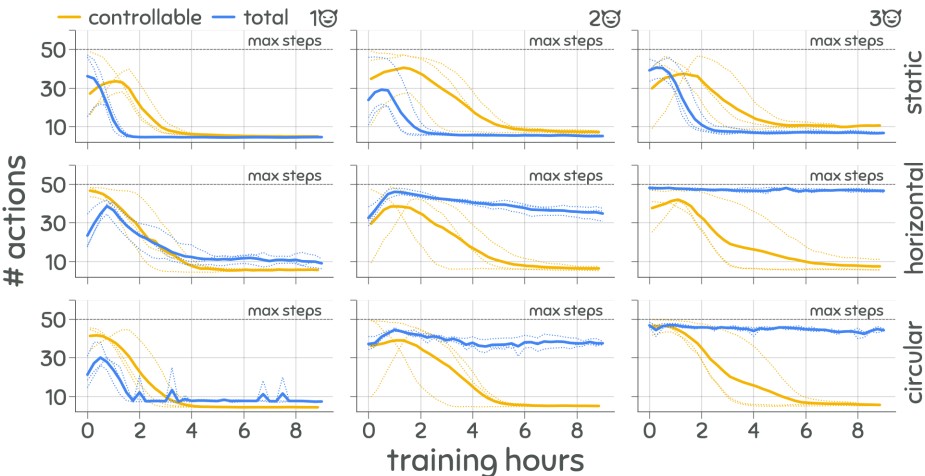

Figure 7: Comparison of the number of actions needed to pick a ball with and without controlled effects. Columns increase the number of demons and rows increase their complexity.

### 4.4. Is there any benefit to use controlled effects?

Throughout this work, we have proposed to disentangle controlled effects from total effects, i.e., we deliberately ignore uncontrolled effects since the agent cannot cause these by definition. The intuition behind this choice is that controlled effects reduce the effect space to model. Disentangling effects is costly; therefore it must be worth doing.

This experiment compares how controlled effects differ from total effects when learning to pick up a ball. The main difference between total and controlled effects is the uncontrolled changes in the environment. Thus, for this experiment we increase the number of demons and the complexity of their dynamics to measure if controlled effects provide a benefit by ignoring the uncontrolled part of the state. We define three different dynamics: **static** where the demon does not move; **horizontal** where demons only move in a horizontal line; and **circular** where the demon describes a circular movement in the arena. Additionally, we do not train the total effects model when using total effects. We report the average number of actions taken to pick the ball.

The first row in Figure 4.4 shows that in the absence of dynamics (static demon), total effects learn more efficiently than controlled effects. This is expected since there is an overhead to train a forward model; moreover the model will not be perfect either. By comparing the use of controlled against total effects column wise, we can see that the performance when using total effects decreases the more demons there are. On the other hand, the agent using controlled effects has similar performance no matter the number of demons used. Furthermore, the agent using total effects does not learn how to pick the ball in the horizontal and circular variants with neither two and three demons.

### 4.5. How does the task policy use the hierarchy of effects?

This experiment evaluates how the task policy uses the hierarchy of effects i.e. does it use abstract effects or does it use effects closer to actions? For this, we use the variant "T" of the environment where the agent needs to go to a target location. This variant can be solved in multiple ways, for

example, the task policy could have learned to use only basic effects to change the $x$ or $y$ coordinates of the agent one step at a time; it could use more complex effects like go-to-target; or a mix of these two. Ideally, the use of higher levels of the hierarchy is preferred since these allow for better credit assignment.

The results in Figure 8 show how the policy learns to alternate between the two high-level effects that lead to complete the task. By working at a higher time scale, together with the environment's reward function, the agent is motivated to use more abstract effects.

## 5. Related Work

**Intrinsic rewards:** a popular solution to the exploration and credit assignment problems is to use intrinsic rewards (Singh et al., 2005; Pathak et al., 2017; Burda et al., 2018; Song et al., 2019; Choi et al., 2019; Badia et al., 2020). This approach promotes exploratory behavior by rewarding curiosity. Burda et al. (2018) use an untrained neural network to estimate surprise and reward for it. Choi et al. (2019) and Badia et al. (2020) reward for the discovery of states controlled by the agent. We consider these methods complementary to our work and can be incorporated to CEHRL for better exploration.

**Hierarchical RL:** Dayan and Hinton (1993) proposed Feudal RL where a hierarchy of managers work at different granularity by controlling the information and rewards transferred to lower levels of the hierarchy. Vezhnevets et al. (2017) gave a deep learning implementation of this framework where a worker performs actions to accomplish the goal set by a manager. Sukhbaatar et al. (2018) on the other hand, divided the training of the manager and worker into exploration and task-dependent stages. These methods rely on extrinsic rewards to learn representations between levels making them ill-suited for environments with sparse rewards due to their high exploratory demands.

The options framework (Sutton et al., 1999) adopts a more decentralized approach to temporal abstraction. Each option represents a skill that decides if it should be started or finished based on the current state. A well-known problem is to discover useful skills (Bacon et al., 2016; Florensa et al., 2017; Eysenbach et al., 2018; Nair et al., 2018; Held et al., 2018; Jegorova et al., 2018; Sharma et al., 2019). A common practice is to optimize the mutual information between skills to make them easily distinguishable. Instead, we consider every way an agent can modify the state as a potential skill.

**Generative RL:** generative models like Variational Autoencoders (VAE) or Generative Adversarial Networks (GAN) have proven to be extremely useful in image or audio generation. Nonetheless, these methods are gaining popularity in RL (Held et al., 2018; Nair et al., 2018; Nair and Finn, 2019). Nair and Finn (2019) used a VAE to produce intermediate goal-images that facilitate the planning of actions to achieve a final goal. Using images as goals in environments with dynamics makes difficult to decide whether a goal has been achieved. Instead, CEHRL removes dynamic effects, only using controlled effects. Additionally, using effects instead of states reduce the size of the distribution to model, e.g. in a 1D infinite environment a generator that works with states would need to produce infinite states to achieve the effect of moving forward. In contrast, CEHRL would generate the same effect over and over. Similarly, Held et al. (2018) and Campero et al. (2021) used GANs to produce a curriculum of goals to ease the learning of a specific task.

**Causality:** Incorporating concepts from causality to deep learning in general (Bengio et al., 2019; Chattopadhyay et al., 2019; Ke et al., 2019) and reinforcement learning in particular (Buesing et al.,

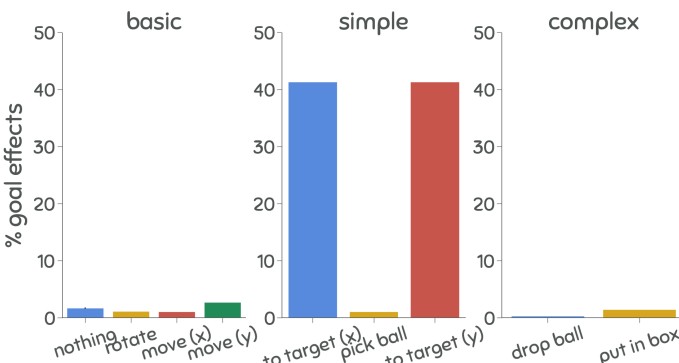

Figure 8: By comparing the three panels, the task policy learns to solve task "T" using abstract effects instead of low-level effects.

2018; Jaques et al., 2018; Dasgupta et al., 2019; Goyal et al., 2019; Nair et al., 2019) has shown to be an important research avenue that can benefit a wide range of tasks. Badia et al. (2020) use an inverse model to identify controlled aspects of the environment. Although efficient, this approach is local to the agent; ignoring changes faraway from the agent. By using counterfactual measures we identify a broad set of controlled effects.

## 6. Conclusion and future work

We presented CEHRL, a hierarchical method that leverages causal tools to continuously learn a hierarchy of controlled effects. We showed that this hierarchy can be used to efficiently learn task-specific behavior in an scalable and efficient manner. Our experiments also show that the cost of building this hierarchy is quickly amortized when the number of tasks or their complexity increases.

To avoid confounding our experiments, CEHRL has been applied to entity-centric representations. In future work, we want to incorporate methods for unsupervised state representation learning (Burgess et al., 2019; Anand et al., 2019) so CEHRL can learn from observations. Additionally, to avoid biasing exploration, effects are sampled uniformly. Instead, we could choose goals from the hierarchy that optimize the learning of novel effects. Another interesting avenue is to use the hierarchy of effects as a temporally abstract model of the world by conditioning it to the state, so only effects achievable in certain number of steps are selected. The possibility of creating social norms ought to be explored using counterfactuals and measures normality in a multi-agent setting.

### Limitations

*Deterministic transitions:* this work, more specifically Eq. 5, considers a deterministic setting. The mode can be approximated for stochastic domains or domain with large number of actions using the mode-collapse problem/feature in GANs.

*Representation learning:* CEHRL works best when the state representation is disentangled and/or compositional. In many cases access to this kind (or any) of state is impossible. Representation learning is an important area of research that we hope to incorporate into CEHRL in the future.

*Normality*: although we propose a measure of normality in Eq. 5, this is far from ideal. We believe the way humans see normality is context dependent and should be learned instead of a fixed function. This is an active research area by psychologist looking to underpin human causal judgment. *Uncontrolled effects*: there may be uncontrolled events required so the agent can fulfill more complex controlled effects, e.g. moving a heavy box with two agents. The high-level policy and prior model do not take this kind of effects into account.

## Acknowledgments

The authors would like to thank Jaan Aru, Daniel Majoral, Ardi Tampuu, Tambet Matiisen, Roman Ring and Meelis Kull for insightful comments on the manuscript. This work was supported by the AWS Cloud Credits for Research program and by the University of Tartu ASTRA Project PER ASPERA, financed by the European Regional Development Fund.

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

## Appendix A. Implementation

Here we describe more details related to the training and random effect exploration of CEHRL and how they are used for task-agnostic and task-specific learning.

### A.1. Implementation details of task-agnostic learning

In this section, we provide additional details to CEHRL's training for task-agnostic learning. To decide when a goal is reached, we define a function $h(e_c, e_c^{\text{goal}})$ that compares if two controlled effects match at each dimension

$$h(e_c, e_c^{\text{goal}}) \equiv \mathbb{1}\left\{e_c = e_c^{\text{goal}}\right\}. \tag{10}$$

**Dealing with effect imbalance:** The replay memory is dominated by basic effects e.g. moving forward happens more often than turning the light on. This makes rare effects difficult to learn. To alleviate this imbalance, each experience is added to the buffer with an initial priority proportional to how rare an effect is

$$g(e_c) = \min\left\{10^5, \frac{1}{p(e_c|\mathcal{D}) + \epsilon}\right\}, \tag{11}$$

where $p(e_c|\mathcal{D})$ is approximated by a running average of the seen effects, and the final value is capped.

**Shared data, individual priorities:** Every component shares the same experiences stored in the replay memory $\mathcal{D}$ but we use an dedicated priority queue for each component. We use the same initial priority but each component updates its priorities based on their individual errors.

**Fixed $\epsilon$-greedy exploration:** The effect-conditioned policy produces a sequence of actions to perform an effect on the environment. This policy relies on $\epsilon$-greedy exploration to find the right action sequence. Typically, the exploration rate $\epsilon$ decays over time. Instead, we adopt an approach similar to Ape-X (Horgan et al., 2018) where this value is fixed throughout the training. Since CEHRL does not work in a distributed manner, we define a set of candidate epsilons $C = \{\epsilon_1, \dots, \epsilon_L\}$ and fix one randomly for each episode. Additionally, we use a warmup period where $C = \{1\}$. This period is used during the initial training of the total effects model and the distribution of effects, see Table 1 for more details. We use a function $f_C \colon t, M \to C$ that decides which epsilons to use based on the current training step and the set of models to be trained.

**Hindsight learning:** To speed up the learning of effects, we augment the collected data in a similar way to Hindsight Experience Replay (HER) by Andrychowicz et al. (2017). Every experience $(s, e_c^{\text{goal}}, a, s', e_c(s, a), r', d')$ is augmented with an additional experience where the goal effect has been replaced with the reached controlled effect i.e. $(s, e_c(s, a), a, s', e_c(s, a), r' = 1, d' = 1)$.

### A.2. Implementation details of task-specific learning

The task policy is trained by performing random effect exploration. This method sets controlled effects as goals where an effect-conditioned policy tries to perform them on the environment. Unfortunately, this last policy will not be perfect and may fail to perform some goals, thus we do not add them to the replay buffer for training.

In contrast to fixed $\epsilon$-greedy, we use the usual $\epsilon$-greedy decay to explore different goal sequences before relying only on the learned value function. Note that even with $\epsilon = 0$ CEHRL still explores the environment, this is possible thanks to random effect exploration.

## Appendix B. Experimental setup

Here we specify the architectures used for CEHRL and the baseline. We also provide the hyperparameters used in our experiments and the search range of hyperparameters done. Furthermore, we provide more specifics on the environment used.

### B.1. Environment

The environment is implemented using MiniGrid 2D by Chevalier-Boisvert et al. (2018). The agent can move forward, turn left or right, pick up different objects, and put objects into boxes. The variants T, BT, and CBT of the environment have all the same number of objects but the task rewarded for is different. Agents are provided with a discrete, entity-centric representation of the world as in Baker et al. (2019). This representation is a $J \times I$ matrix with the $J$ objects in the arena (including the agent) and the following $I$ attributes per object: type of object, x and y coordinates, color for objects or direction for the agent, and the carrying object. If the agent achieves a task, the environment gives a reward using the following function

$$r = \begin{cases} 1 - \frac{0.9 * t}{T} & \text{solved} \\ 0 & \text{otherwise} \end{cases} \tag{12}$$

where $t$ is the current time step and $T$ is the maximum time steps per episode.

### B.2. Architecture

We implement every network as an MLP with fully connected layers and ReLU activation functions.

**State encoder:** We use a network common to other components to encode the state. Each attribute type in the state is encoded into the same embedding space. We concatenate the resulting embeddings, and process them with a two fully connected layers to create a reduced latent space. See Figure 9.

**Effect encoder:** Similarly, the effect encoder creates a low-dimensional latent space by using the effect as input of two fully connected layers. See Figure 10.

**Total effects model:** To predict the dynamics of the environment, we provide this component with two consecutive states. We implement this component with two state encoders to encode each consecutive state into a latent vector. These two latent vectors are then processed by four fully connected layers to output the predicted total effects for each of the actions. See Figure 11.

**Distribution of effects:** This component is implemented as a VAE with four fully connected layers to encode controlled effects into a latent space and three fully connected layers to decode latent vectors to controlled effects. Note that the last layer of the decoder has a Sigmoid activation function so as to output a binary vector. Consequently, we replace the typical mean-squared-error reconstruction loss of the VAE with a binary cross-entropy loss. See Figure 12.

**Effect-conditioned Q-value function:** This component uses state and effect encoders to create a latent representation of its input. This latent representation is then passed to a four fully connected layer network with an additional Siamese layer to compute state-value and advantage functions, as used in dueling networks. See Figure 13.

**Task Q-value function:** This is the same network as the effect-conditioned Q-value function but without the Siamese network since we do not have a limited set of controlled effects. See Figure 14.

**Baseline**: We use the same network as the effect-conditioned Q-value network but without the effect encoder, and train it using DQN with prioritized experience replay, double DQN and dueling networks. This baseline uses the fixed $\epsilon$-greedy exploration described above. Since the baseline is composed of only a policy, we do not do any warmup phase.

### B.3. Hyperparameters

Table 1 provides the training schedule used to implement function $f_M$ and $f_C$. Hyperparameters common to every training and models are shown in Table 2. Hyperparameters for task-agnostic learning are shown in Table 3. Hyperparameters for task-specific learning are shown in Table 4. Hyperparameters for the baseline are shown in Table 5. Note that each table also provides the search range we used to fix each hyperparameter, if applicable.

| Models | Steps | Use warmup $C$ |
|---|---|---|
| $\{\hat{e}_t\}$ | [90000, 0] | True |
| $\{\hat{e}_t, E\}$ | [30000, 0] | True |
| $\{\hat{e}_t, \pi_e\}$ | [30000, 0] | False |
| $\{\hat{e}_t, E\}$ | [10000] | False |
| $\{\pi_e\}$ | [60000] | False |

Table 1: Training schedule used by $f_M$ and $f_C$. Every $X$ steps the function $f_M$ selects the next row in the scheduling table. Once the function reaches the end, it starts again from the first row on the second (or last) Steps field.

| Hyperparameter | Value | Search range |
|---|---|---|
| Batch size | 128 | (32,64,128,256) |
| Priority replay buffer $\alpha$ | 1.0 | N/A |
| Priority replay buffer $\beta$ | 0.01 | N/A |
| Discount factor | 0.85 | (0.8,0.85,0.9,0.95,0.99) |
| Training frequency | 5 | N/A |
| $N$ | 20 | N/A |
| $P$ | -0.02 | N/A |
| $K$ | 50 | N/A |

Table 2: Common hyperparameters.

| Hyperparameter | Value | Search range |
|---|---|---|
| Task-agnostic replay capacity | 500K | (200K,500K,1M) |
| C[warmup] | $\{1\}$ | N/A |
| C | $\{0.8, 0.6, 0.4, 0.2, 0.1, 0.01\}$ | N/A |
| State encoder units | 128 | (64,128,256) |
| State encoder latent | 32 | (8,12,16,32) |
| Effect encoder units | 256 | (64,128,256) |
| Effect encoder latent | 12 | (8,12,16,32) |
| $Q_e$ units | 512 | (64,128,256,512,1024) |
| $Q_e$ learning rate | 0.0001 | (1e-3,5e-4,1e-4,5e-5,1e-5) |
| $Q_e$ target update | 15K | (1K,5K,10K,15K,20K) |
| $E$ encoder units | (256-128-64) | (512,256,128) |
| $E$ decoder units | (64-128-256) | (512,256,128) |
| $E$ latent | 8 | (4,8,16,32) |
| $E$ learning rate | 0.001 | (5e-3,1e-3,5e-4,1e-4,5e-5) |
| $\hat{e}_t$ learning rate | 0.0005 | (1e-3,5e-4,1e-4) |
| $\hat{e}_t$ units | 32 | (32,64,128,256) |

Table 3: Hyperparameters for task-agnostic training.

| Hyperparameter | Value | Search range |
|---|---|---|
| Task-specific replay capacity | 100K | N/A |
| $Q_t$ units | 32 | (8,16,32,64,128,256) |
| $Q_t$ learning rate | 0.001 | (1e-3,1e-4,5e-4,5e-5) |
| $Q_t$ target update | 2K | (1K,2K,5K,10K,15K) |
| $Q_t$ epsilon start | 1.0 | N/A |
| $Q_t$ epsilon end | 0.0 | N/A |
| $Q_t$ epsilon steps | 50K | N/A |

Table 4: Hyperparameters for task-specific training.

| Hyperparameter | Value | Search range |
|---|---|---|
| Replay capacity | 500K | N/A |
| C | $\{0.8, 0.6, 0.4, 0.2, 0.1, 0.01\}$ | N/A |
| State encoder units | 256 | N/A |
| State encoder latent | 12 | N/A |
| Effect encoder units | 256 | N/A |
| Effect encoder latent | 12 | N/A |
| $Q_b$ units | 256 | (32,64,128,256,512) |
| $Q_b$ learning rate | 0.00005 | (1e-3,1e-4,5e-4,5e-5) |
| $Q_b$ target update | 15000 | (1K,5K,10K,15K,20K) |

Table 5: Hyperparameters for the baseline.

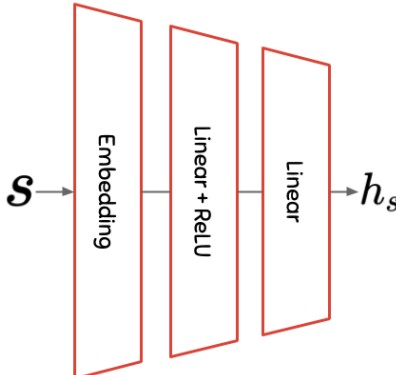

Figure 9: State encoder architecture

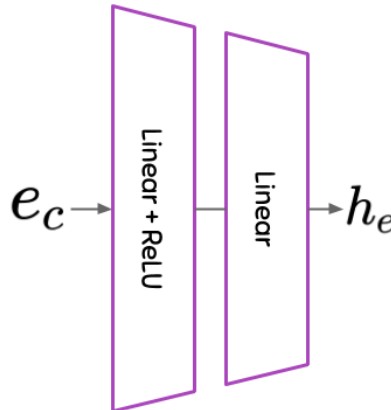

Figure 10: Effect encoder architecture

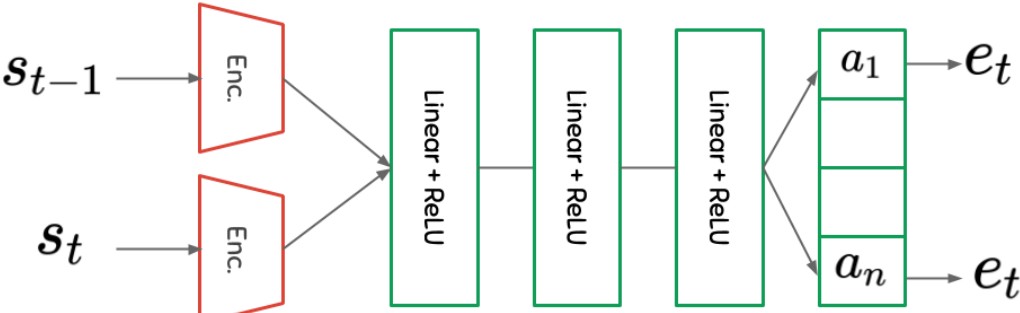

Figure 11: Total effects model architecture

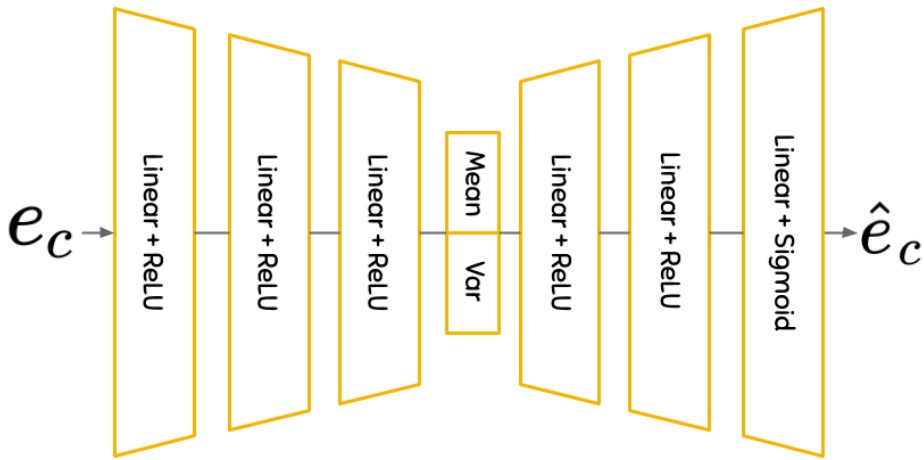

Figure 12: VAE architecture

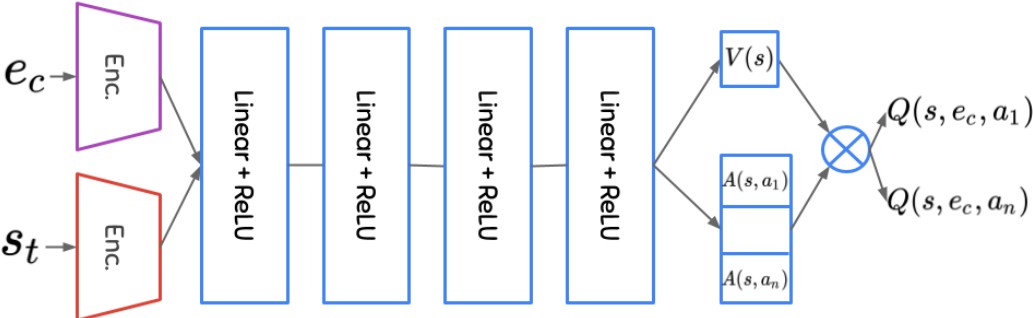

Figure 13: Effect-conditioned policy's q-value architecture

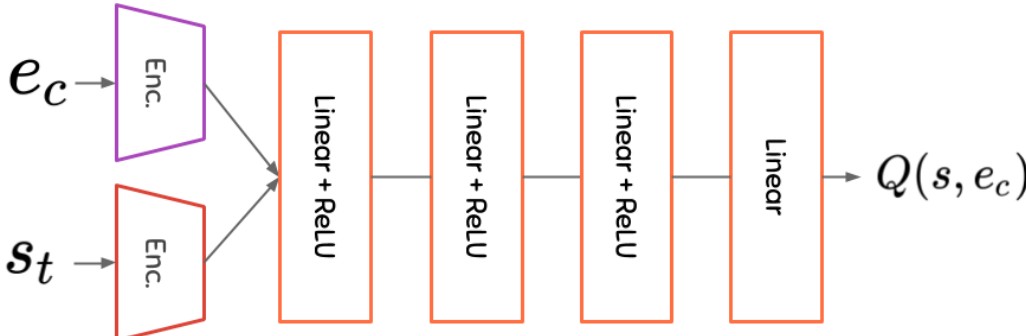

Figure 14: Task policy's q-value network architecture for task-specific learning

