# OpenReview forum: "Disentangling Controlled Effects for Hierarchical Reinforcement Learning"
_cclear.cc/CLeaR/2022/Conference — CLeaR 2022 Poster_

### Official Review · Reviewer_wXCW · 2021-11-11

**Confidence:** 3
**Overall Score:** 5

**Main Review:**

The paper proposes an interesting hierarchical RL algorithm, that attempts to decompose a long-horizon sparse reward task into compositional sub-tasks. These are accomplished using (1) a task-specifying meta-controller; (2) a VAE to sample feasible tasks; (3) a task-conditioned low-level action policy. The only causal connection is in identifying the effect of an action -- this is done using essentially advantage functions, which are re-interpreted as causal effect estimates. There are other approaches in the literature that attempt to isolate the causal effect of actions from uncontrollable effects (e.g., the literature on intrinsic motivation, curiosity-driven exploration, solutions to the "Noisy-TV problem", or exogenous decompositions as in http://proceedings.mlr.press/v80/dietterich18a/dietterich18a.pdf) that are not discussed or compared to. The review below focuses only on the weaknesses of the paper.

- Please clarify that s is a p-dimensional vector, transitions are deterministic, and actions are discrete (before defining e_t(s,a) = s' - s). In Eqn5, mode_a e_t(s,a) is confusing -- how are ties broken? Why is this a reasonable "normative world"? What are the properties needed of beta_Y the normative world, and how does mode_a ... satisfy those properties?

- Eqn 3: Why is V(s) a reasonable choice for a normative world G_{tilde a}? V(s) depends on an agent's policy, while G_{tilde a} and beta_Y appear to be defined independent of policy. Can you please clarify why advantages can be viewed as causal effects (rather than the conventional view that they are variance-reduced estimates of cumulative returns Q(s,a))?

- Better figure: Fig2 helps to understand the main pieces of CEHRL, but does not spell out how it connects to Alg 1 and 2. Rather than Fig3, I would appreciate a figure that clarifies: two different timescales -- per-environment step and per-effect step, the timescales of experiences to train hat{e}_t (assuming it is per-environment step), E (per-effect step), pi_t (per-effect step), pi_e (per-environment step); switching from exploration phase to task-specific learning phase.

- Why does pi_t need e_c as input? Is identifying the next task a non-Markovian problem, and does e_c somehow convey sufficient information to make pi_t learnable?

- Eqn 5, 6: This approach of disentangling controlled effect e_c from total effect e_t requires more discussion. Prior work (e.g., https://pathak22.github.io/noreward-rl/) uses a state embedding s -> z which attempts to capture the controllable state factors. This is typically accomplished by looking at what aspects of the s-s' transition are predictive of the action taken. What are the benefits/drawbacks of learning a hat{e}_t model (which will suffer aleatoric uncertainty from uncontrollable factors) and then subtracting the mode (hoping that the uncontrollable factors affect both e_t and the mode in the same additive way); as opposed to learning a controllable minimal latent state representation?

- [Minor clarification] Mismatch between definition and use of e_c^goal: e_c^goal is sampled from E which is a VAE trained on all (s,a) pairs in the experience replay. e_c(s,a) from Eqn 5 gives per-environment-step controlled effect estimate. However, it is being used as a multi-step goal in Alg 1 (e_c^goal in line 5 is uniformly sampled from E and relates a goal state to s_0; has_achieved however compares e_c step which relates s_current-s_prev to e_c^goal which is s_desired - s_0?) Why is this "multi-step" usage of e_c^goal consistent with its one-environment-step definition?

- Clarify in Alg 1 pseudocode that pi_t is uniform (Eqn 8).

- Alg 2: Is the r' <- r' * (1-d) update supposed to happen before the "if h then" block? Also should it be r' * (1-h) [strange that we are rewarding a trajectory even when it terminates because t > K].

- (See comments above for other baselines to use in experiments). The approach of Badia et al 2020 seems especially important to benchmark against.

- Strange that the example using in Fig 1 is not revisited in the minigrid2D domain and tested empirically.

- Please discuss learnability issues: (this is hinted in the last paragraph about conditioning E such that only effects achievable within K are selected) An implicit assumption of Alg 1 and 2 is that pi_e can be learned for all the goals that E could emit. Why is this a reasonable assumption? How delicate is CEHRL to such an assumption?

**Summary:**

Hierarchical RL algorithm with very little insights or contributions to causal learning

---

> ### Author Response · Authors · 2021-12-02
> **Response to reviewer wXCW**
>
> > Please clarify that s is a p-dimensional vector, transitions are deterministic, and actions are discrete (before defining e_t(s,a) = s' - s).
>
> We agree that this information is better suited before defining $e_t$ and not after. We will change this in the manuscript.
>
> > In Eqn5, mode_a e_t(s,a) is confusing...
>
> Although this work explores a fixed rule (the mode) to create a normative world, this does not need to be the case. Presumably, a learned context-dependent rule can model better what is normal; possibly closer to a human measure of normality. More research needs to be done to explore how normality can be made more flexible. Indeed, this is an active research topic in psychology. We will add this discussion to a new "limitations" section.
>
> > Eqn 3: Why is V(s) a reasonable choice for a normative world G_{tilde a}?...
>
> The advantage function computes how much the return would change with respect to a policy if we were to take action $a$ instead of another action, i.e., how much improvement the agent gets from changing its policy. Eq. 2 does not impose any restriction on what kind of normal world to use. In this case, the policy is taken into account since it is easier to estimate.
>
> An interesting question is if there is a benefit in considering each action equally (not policy weighting at the first level of the value function) when computing the normal world. Note that this question is harder to answer since we must have a good estimate for each action independently of how often the policy uses it. Since the advantage function is typically used to improve the policy itself, there may not be many benefits from doing this, but it may in other contexts/applications.
>
> > Why does pi_t need e_c as input?...
>
> * Just to clarify, $pi_t$ is not conditioned on a single effect but multiple candidates generated by the VAE model. We describe this in section 3, but we do agree that the bold style of the candidate effects $\boldsymbol{e_c}$ can be missed.
> * The generative model $E$ encloses the dynamics of the world and is responsible for generating plausible candidates that are task-independent. Then, $\pi_t$ chooses which candidate to perform in the context of a specific task. This way, we can learn the expensive model $E$ once without rewards. We leave for future work the task of making the generative model take reward into account without the cost of training it for each task.
>
> > Eqn 5, 6: This approach of disentangling controlled effect e_c from total effect e_t requires more discussion...
>
> Inverse models have indeed been used to identify controllable elements of the environment. Although a careful study is needed to compare these methods, we argue that these methods suffer in two cases. 1) When multiple actions map to the same effect. For example, if an agent moves to a wall or takes the "do-nothing" action, the model will not predict the action accurately, and consequently, its representation will most probably suffer.
> 2) An inverse model does not promote modeling all controlled effects. For example, if the agent moves a box when moving left, the inverse model only needs to encode the agent's movement to predict the action accurately and can freely ignore the box's movement.
>
> In contrast, Eq. 5 encourages to model every change in the scene by using a forward model. Additionally, our method identifies controlled effects at pixel level instead of latent representations, which can benefit interpretability and many other applications. Of course, this comes with the cost of training a forward model. We will add an overview of these methods to the related work section.
>
> > [Minor clarification] Mismatch between definition and use of e_c^goal...
>
> * Just to clarify, the VAE $(E)$ is trained on controlled effects $e_c$. This is explained in section 3.1 (learning). The idea is that $E$ will generate different ways the agent can control the environment. Note that $E$ is not conditioned on the state.
> * As mentioned by the reviewer, the generative model produces temporally abstract goals (multi-step goals). Note that since the VAE is trained on any discovered controlled effect and is not conditioned on the state, $E$ produces changes agnostic to the time step. In other words, "has_achieved" does not compare s_desired - s_0 but to e_c^goal, which is not fixed to a time step. For example, if the agent (on the exploration phase) reaches the state $s'=[0, 0, 1]$ from the state $s=[1, 0, 0]$ and let us say that $e_c = [-1, 0, 1]$. The goal set by $E$ will be $e_c^\text{goal}=[-1, 0, 1]$ irrespective of the current time step the agent is at right now. So the has_achieved comparison uses $e_c(s_t, a_t)$ and this generic goal $e_c^\text{goal}$. The reason to do this is that both effects are under the agent's control and can be easily compared, in contrast to using the total effects with controlled and uncontrolled changes entangled.

---

> > ### Author Response · Authors · 2021-12-02
> > **continuation to response**
> >
> > > Better figure: Fig2 helps to understand the main pieces of CEHRL, but...
> >
> > We agree that a figure showing lower-level mechanics of the model would be beneficial to the reader. We will add this figure to the manuscript.
> >
> > > Clarify in Alg 1 pseudocode that pi_t is uniform (Eqn 8).
> >
> > True, we will correct this.
> >
> > > Alg 2: Is the r' <- r' * (1-d) update supposed to happen before the "if h then" block...
> >
> > It should happen after. This line in Alg. 2 accumulates the reward given at each environment step into a single reward for each effect step. The termination of an effect step is always determined by when a goal terminates either because it was accomplished or because it exceeded the time limit or because the episode finished.
> >
> > > The approach of Badia et al 2020 seems especially important to benchmark against.
> >
> > In this work, we decided to use DQN as a baseline to understand the benefits of temporal abstraction using counterfactuals. We recognize that an inverse model in addition to our current baseline would indeed give a broader context to the reader.
> >
> > > Strange that the example using in Fig 1 is not revisited in the minigrid2D domain and tested empirically.
> >
> > We concur that this would add continuity to the reader. There is no particular reason not to have this environment. We chose tasks where we could test agents at different levels of difficulty and dynamics.
> >
> > > Please discuss learnability issues...
> >
> > We will add a new "limitations" section with the following concerns (in addition to suggestions by other reviewers):
> >
> > * Although Eq. 5 identifies goals that the agent controls, there may still be uncontrolled events required to happen so the agent can fulfill this goal. For example, moving a heavy box with two agents.
> > * Eq. 5 uses the mode as the normative world, but this may not be ideal in some cases. More research needs to be done to find better ways to compute normative worlds.
> > * CEHRL works with the state, which can be inaccessible in complex environments.

---

### Official Review · Reviewer_B1ya · 2021-11-22

**Confidence:** 4
**Overall Score:** 4

**Main Review:**

### Originality

The proposed method seems original to me.

### Significance

The paper is not particularly relevant or significant to the CLeaR community, even if it tries to address an important problem, that of exploration and scalability in RL. While the authors try to relate their method to existing concepts in causality, I feel that the connection is very thin, and not really justified. In the end the proposed method is very similar to model-based RL, where the transition model is learned by residual learning ($f(s'-g(s'|s)|s,a)$ instead of $f(s'|s,a)$), and where the residues are used as goals for exploration. The proposed method might be interesting to the (hierarchical) RL community, but a comparison (methodological and experimental) to existing hierarchical RL methods is missing to judge of the significance of the proposed method.

### Technical quality

The paper is not technically sound. Several concepts appear to be ill-defined (see detailed comments), the assumptions are not clearly stated from the start, the experimental setup appears to be flawed (comparing a method with pre-training to a method learning from scratch) or irrelevant (comparing a learned/learning policy to a random policy), and the paper is poorly written in general (it is still unclear to me how the proposed method works, which parts of the bells-and-whistles are required for it to work, if those bells-and-whistles are also part of the baseline method, and finally what exactly is the experimental setup).

### Clarity

I found the paper to be poorly written overall, although it is well-written in terms of language. It is hard to parse what the core contribution of the paper is, and what are just implementation details thrown here and there (see detailed comments). I also found the motivation not well explained. How does eq.5 disentangle effects from the agents ?

### Detailed comments

p.1: we propose agents that learn by estimating the causal effect of an action on the
state -> Isn't that what model-based RL do already ? How does your approach differ from model-based RL ?

p.2 Fig.1: I would suggest improving that Figure and its caption. It took me q while to understand how (a) and (b) are related, and simply to understand what this figure describes (an RL environment).

p.3 Fig.2: How different is that from hierarchical RL ?

p.3: the definition for $e_t(s,a)$ seems rather abstract, and is not mathematically correct since $s'$ appears only one side. Unless you only consider deterministic environments where $s'=f(s,a)$ ? Also, what about state spaces that are not equipped with addition ? Or state spaces where addition does not make sense ?

p.3 eq.1: Here again, $\hat{x}$ only appears on the right-hand side.

p.4 eq. 5: $\text{mode}_{a_i \in A} e_t(s, a_i)$ -> this does not make sense. from what I understand $e_t$ has same dimensionality as the state space, and therefore can be multi-dimensional. How then do you decide on the mode of a multi-dimensional function ?

p.4 eq.6: $e_t$ is missing arguments here. I suppose you mean $e_t(s,a)$ ? Then you're basically learning a model of the world, via residual regression.

p.4: In our experiments, we assume [...] -> You should make it clear from the beginning what are the assumptions required by your method.

p.5: distribution of effects E -> is this distribution conditioned on a state $s$ ? I don't see how an effect makes any sense without the corresponding original state.

p.5: we store experiences of the form [...] -> why storing $e_c(s,a)$, when it can be deduced from $s'-s$ ? Also, $r'$ and $d'$ haven't been defined yet.

p.5: or a punishment $P$ otherwise -> why adding a punishment ? Is a necessary trick for the approach to work ? Or is there something more fundamental ?

p.5: Experiences are augmented [...] to each experience. -> These sound like a lot of bells and whistles. How is each of these components related to your contribution ? Are they necessary for your approach to work ? Are the observed experimental improvements due to these components, or to the fact that you model causal effects ? I suggest to either remove these components altogether, or perform a proper ablation study with those components on and off to help the reader in performing a proper credit assignment.

p.5: Distribution of effects -> I do not understand how this relates to your approach as described in Section 2, or how modeling a distribution of controlled effects $s'-\hat{s}'$ without conditioning on a state and action makes sense at all.

p.5 $1: $(e_c^{goal}|s,e_c)$ what is $e_c$ here ? Does it come from a specific action ? Which one ?

p.6: Task policy -> same comment here. I do not see how aiming for a change $s'-s$ hat does not consider the current stats $s$ makes any sense at all. Why not just learning a distribution over states then, and set those states as goals ? And why do you keep $s$ in the definition of $\pi_t$ on the left-hand-side, if in the end you don't use $s$ ?

p.6 alg.1: What do add_with_her or train_if_needed do ? Those routines are not defined.

p.7: CEHRL and the baseline -> what is the baseline ? It's not been described yet.

p.7: using the pre-trained models -> it seems your approach then is cheating ? Does the DQN baseline also have access to the samples you use for pre-training your models ?

p.8 Fig.4: the x-axis should report the number of interactions with the environment, not the training time, to measure sample-efficiency without bias.

p.8: random effect exploration performs [...] three times more often -> this is very much expected, if you compare a random policy to any policy that has learned a little bit on a non-trivial task, the random policy will achieve less complex task more often.

p8: we record [...] by each method -> which methods ? Do you record training samples ? Does your method start from scratch, or does it use pre-trained models ?

p.9: controlled effects reduce the effect space to model -> I don't see how. If I expand the controlled effect definition, I get $s'-\hat{s}$. How is the resulting space smaller than that of the total effect $s'-s$ ?

**Summary:**

The paper proposes a new hierarchical RL method, which uses finite differences in the state space (so-called effects) as sub-goals for the training agent. The authors propose an implementation of their method, and present a series of experiments in a grid world setting.

---

> ### Author Response · Authors · 2021-12-02
> **Response to reviewer B1ya**
>
> > the proposed method is very similar to model-based RL...
>
> CEHRL is a model-based RL algorithm where a generative model $E$ and a low-level policy $\pi_e$ enable temporal abstraction. A necessary clarification is that CEHRL does not use residues/total effects as goals but controlled effects. This is stated throughout the paper; moreover, section 2 is devoted to identifying these controlled effects. Note that the goal space is largely reduced by using controlled effects instead of total effects/residues as goals. We provide evidence for this in experiment 4.4.
>
> > How does eq.5 disentangle effects from the agents?
>
> Section 2 describes this in detail. In essence, Eq. 5 compares what happened with a normative world to determine if the agent controlled an effect (change on the state) or not.
>
> > p.1: Isn't that what model-based RL do already?...
>
> CEHRL falls under the model-based paradigm where a generative model $E$ and a low-level policy $\pi_e$ enable temporal abstraction. CEHRL differentiates from other model-based approaches by 1) creating a world model that considers only controlled aspects of the environment using counterfactual measures; 2) enabling temporal abstraction by combining a generative model and a low-level policy.
>
> > p.3 Fig.2: How different is that from hierarchical RL?
>
> We propose a hierarchical RL (HRL) method and do not claim that we are doing something different from HRL. There are different approaches to hierarchical RL; Fig. 2 reflects our approach.
>
> > p.3: the definition for $e_t(s,a)$ seems rather abstract, and is not mathematically correct since...
>
> We indeed consider a deterministic setting. For stochastic environments, the mode should include all possible next states. We will add this limitation and ways to solve it in a new "limitations" section. The key idea to estimate the mode for stochastic domains (and to avoid a call per action to the NN) is to use something similar to the mode-collapse problem/feature in GANs.
>
> > p.4 eq. 5: $\text{mode}_{a_i \in A} e_t(s,a_i)$-> this does not make sense...
>
> The mode is computed per dimension. We will add a clarification to section 2.
>
> > p.4 eq.6: $e_t$ is missing arguments here...
>
> Thanks, we will change to $e_t(s,a)$. Indeed, we use the total effect as target of the forward model $\hat{e_t}$.
>
> > p.4: In our experiments, we assume [...] -> You should make it clear from the beginning...
>
> Indeed, we will move these to the definition of $e_t$.
>
> > p.5: distribution of effects E -> is this distribution conditioned on a state?...
>
> The generative model $E$ is not conditioned on the state. By doing this, CEHRL considers all possible controlled effects as goals. Note the generative model produces controlled effects achievable in K steps and not just the next/single step. Including the state would, in many scenarios, reduce the complexity of the goal-space. We decided to leave this addition to future iterations of the proposed framework.
>
> > p.5: we store experiences of the form [...] -> why storing, when it can be deduced...
>
> It is true that $e_c$ is redundant information and can be deduced from s and s'. We will correct this in the manuscript and move the definition of d' and r'.
>
> > p.5: or a punishment other wise -> why adding a punishment ?...
>
> Adding a small punishment is a common approach to motivate the agent to complete goals as soon as possible. It is not a requirement for the approach but a desired behavior.
>
> > p.5: Experiences are augmented [...] to each experience. -> These sound like a lot of bells and whistles...
>
> Augmenting experiences, as in Hindsight Experience Replay, is a popular approach to learning goal-conditioned policies.
>
> > p.5: Distribution of effects -> I do not understand how this relates to your approach...
>
> The goal is to have agents that operate at a different time scale as the environment (section 3 "Eq. 5 can be used for temporal abstraction"). CEHRL does not generate the next (one time step) controlled state but any possible future (K time steps) controlled by the agent. Consequently, conditioning on the action would not achieve temporal abstraction since it would make the agent myopic to a single time step.
>
> > p.5 (e_c^{goal}|s,e_c)e_c$ here ?...
>
> There seems to be a misunderstanding here. As explained before, the generative model produces a set of temporally abstract goals. Here $e_c$ is a possible way of controlling the environment, learned by $E$ using Eq. 5.
> For example, in Fig 1(b) $E$ could generate the candidate goals of "move red box left" or "move agent right" (among others). Note that there is no action "move red box", just move up, down, left and right. These candidate effects are temporally abstract; they could be completed in <K time steps (K actions) and not the next step.

---

> > ### Author Response · Authors · 2021-12-02
> > **continuation of response**
> >
> > > p.6: Task policy -> same comment here...
> >
> > * As explained before, CEHRL works with controlled effects. This approach is considerably different from working with random states.
> > * CEHRL has three main components: $E$, $\pi_e$ and $\pi_t$. $E$ and $\pi_e$ have the same implementation during exploration and task-specific phases. In contrast, $\pi_t$ is conditioned on the state and candidate effects; it just happens that one implementation of $\pi_t$ uses the state (during task-specific learning) and one does not (during exploration). Furthermore, future implementations of $\pi_t$ for exploration may use the state, but the proposed framework stays the same.
> >
> > > p.7: CEHRL and the baseline -> what is the baseline...
> >
> > True, we will move the baseline description to the first paragraph.
> >
> > > p.7: using the pre-trained models -> it seems your approach then is cheating?...
> >
> > No, we are not cheating. Experiment 4.1 studies if having a hierarchy of effects benefits learning new tasks. As stated in experiment 4.3, the baseline took 5 hours to learn two of the three given tasks, and it was given at least 5 hours of training time for the third unsolved task. CEHRL learns the hierarchy of effects in 6 hours and takes ~1 hour to learn the three given tasks. This shows that 1) we have given the baseline even more data than CEHRL and 2) building a hierarchy of effects may seem costly a priory, but it enables the agent to reuse prior knowledge, and consequently, its cost is amortized the more knowledge is reused between tasks. To summarize, the whole point of these two experiments is to show that controlled effects can enable efficient hierarchical learning and, as a consequence, agents reuse prior knowledge to scale better with the number and complexity of the given tasks.
> >
> > > p.8 Fig.4: the x-axis should report the number of interactions with the environment...
> >
> > As mentioned in the paper, it would be unfair to the baseline to report interactions with the environment. DQN can do more interactions per hour due to its shorter training steps (it just trains a single neural net). In other words, DQN may take 1 hour to do 1M steps but CEHRL may take 3 hours to do the same steps since it needs to call to multiple NNs (for inference or training). This would not be reflected when using interactions. Although the numbers would look better for CEHRL when using interactions, we think it is fairer to use hours in this case.
> >
> > > p.8: random effect exploration performs [...] three times more often -> this is very much expected...
> >
> > Experiment 4.2 shows that random effect exploration can explore better even without seeing any task. The generative model and effect policy are trained in an unsupervised RL manner, i.e., even without seeing any biased reward (extrinsic or intrinsic), the agent can learn meaningful exploration. We do agree that it is expected for the random effect exploration to perform better than random actions, but the central message of this experiment is that leveraging prior knowledge is important not just to learn new tasks but to explore and discover novel effects (which in turn helps to learn new tasks).
> >
> > All in all, the message we want to deliver in this work is that, as with humans, controlled effects can be used to acquire prior knowledge and that this prior knowledge is essential for efficient learning.
> >
> > > p.9: controlled effects reduce the effect space to model -> I don't see how...
> > This question seems to be rooted in the same misunderstanding mentioned in a previous question; see our previous clarification.
> >
> > Experiment 4.4 shows that when using total effects, the model does not scale well with the uncontrolled dynamics of the environment but it does when using controlled effects. This is because many total effects will be clustered together by $e_c(s, a)$. For example, under total effects, the effect of the agent moving right when a demon moves left is different than when a demon moves right. In contrast, both cases are the same under controlled effects since the agent does not control the demon. This is a simple case, but since we do not control most changes in our environment (e.g., wind, leaves, cars, rain), identifying controlled effects can lead to considerable gains in efficiency.

---

### Official Review · Reviewer_n7SG · 2021-11-24

**Confidence:** 4
**Overall Score:** 6

**Main Review:**

The paper is full of new interesting ideas. In particular, that of using and learning effects rather than coarser level states or options for HRL. All of the paper, however, relies on a precise definition of effects: mode normalized *differences between states* after taking actions.

A big criticism of mine is that the algorithm in the tabular setting is essentially equivalent to using (random) states as goals, which is nothing new. In a deterministic tabular MDP, if s is a one-hot encoding description of the state, then e_t(s, a) = s' - s is just a +1 on s' and -1 on s. After mode-normalization, e_c(s, a) is +1 in s' and -1 in the s'(s, mode_a). Thus there are at least as many e_c as possible s', and thus the whole thing is essentially a UVFA with random states as goals.

The same criticism applies for high dimensional deep RL settings where we assume s' are essentially one to one with (s, a) pairs for non-terminal s. In this case we typically have a very bad reachability problem of using states as goals since for a given s in general it's impossible to reach a preselected s' (think of a video where we ask that we reach an exact image, this will generally be impossible since we can't control every pixel intensity that may change with the time of day / light / slight changes in motion).

Thus, the algorithm only makes sense for cases where we have a highly descriptive compositional state representation like (coordinate x, coordinate y, do I have the ball) and so forth. This means that for deep RL we are ignoring the representation learning issue, which is OK for a first paper on the subject. However, the fact that it doesn't work at all for tabular cases is in my opinion a bad omen.

I also think the authors should provide themselves this kind of analysis in the paper and be more honest about the limitations of their method.

Other fundamental criticisms I have are as follows:
 - since the e_t used are estimated by a neural network forward model, this is going to be incredibly hard to scale. Learning transition functions that are relevant is difficult in deep RL and typically miss very important information. See for instance https://arxiv.org/pdf/1507.08750.pdf and all this line of research.
- e_c^goals come from a vae which means they are learned vectors. How on earth do they match *exactly* the e_c^step that's used to match in algorithms 1 and 2. This seems very fishy and quite a limitation of the method.

All in all, the paper is interesting and has a lot of new ideas, is clear, and I find the work significant. However, it has a lot of very strong problems that are not even adressed properly in the paper. If they provide a more proper analysis of the limitations in the paper I will increase my score.

**Summary:**

This paper proposes an hierarchical reinforcement learning algorithm based on discovering effects (mode normalized state space differences after acting) and using them to solve tasks.

---

> ### Author Response · Authors · 2021-12-02
> **Response to reviewer n7SG**
>
> > A big criticism of mine is that the algorithm in the tabular setting is essentially equivalent to using (random) states as goals, which is nothing new...
>
> This is an interesting case. If the state is a one-hot encoding, the change in state will, as the reviewer said, be -1 and +1 on s and s' respectively. The change on s will be identified as not controlled since its normative state is -1 (the agent leaves that state no matter what action it takes). Note that the agent may or may not control the change in s'; this depends on the environment dynamics and the agent's action. A peculiar case is when the agent typically would go to $\hat{s'}$ i.e. the mode is $\hat{s'} - s = [-1, 1, 0]$ but taking an unusual action would lead to $[-1, 0, 1]$. In this case, the agent had an effect on two states $e_c(s,a) = [-1, 0, 1] - [-1, 1, 0] = [0, -1, 1]$; that is, the agent avoided one state and made another state happen. This is consistent with, for example, the agent's oxygen decreasing by a constant value when it moves but stays the same when it does not.
>
> In general, there seems to be a slight misunderstanding. The normative world computed in Eq. 5 (the mode) considers the current state, and thus, it is not the "overall" normal state of the system. In other words, the mode is only over actions and not actions and states. We will clarify this in the paper.
>
> Regarding the method being equivalent to using random states as goals. Let us take an extreme case to show that CEHRL is doing something different. If, for example, we have an environment where the agent's actions do not have any effect on the environment, Eq. 5 would output zeros everywhere since the outcome of every action is the same, the mode; even if the environment has infinite states. Thus, for CEHRL there would be only one goal but infinite for the case of the random states. More specifically, CEHRL only considers controlled states and randomly sets one of those states as a goal.
>
> > The same criticism applies for high dimensional deep RL settings where we assume s' are essentially one to one with (s, a) pairs for non-terminal s...
>
> * We indeed consider a deterministic setting. For stochastic environments, the mode should include all possible next states. We will add this limitation and ways to solve it in a "limitations" section. The key idea to estimate the mode for stochastic domains (and to avoid a call per action to the NN) is to use something similar to the mode-collapse problem/feature in GANs.
> * As in our previous answer, Eq. 5 identifies controlled changes only.  Using this kind of goal instead of the entire observation/state, CEHRL alleviates the reachability problem. An example of this benefit is presented in experiments 4.4 where using the whole state instead of controlled effects leads to worse learning scalability the more dynamics in the environment. Nonetheless, we do agree that more work needs to be done to set fully achievable goals.
>
> > Thus, the algorithm only makes sense for cases where we have a highly descriptive compositional state representation like (coordinate x, coordinate y, do I have the ball) and so forth...
>
> Although CEHRL is not limited to compositional states (as described in previous examples), we do agree that our method will work best with highly disentangled states. This, as the reviewer mentioned, is a representation learning problem that we did not consider in this work. We will make this assumption more evident in a new "limitations" section.
>
> > since the e_t used are estimated by a neural network forward model, this is going to be incredibly hard to scale. Learning transition functions that are relevant is difficult in deep RL and typically miss very important information. See for instance https://arxiv.org/pdf/1507.08750.pdf and all this line of research.
>
> Indeed, predicting the next state is extremely hard, but this limitation is not exclusive to our method, and future advances in this area can be incorporated. It is important to notice that although we use a forward model ($\hat{e_t}(s,a)$), it is OK for this model to not be perfect on uncontrolled effects. The only requirement is that it makes the same mistakes in every alternative world, e.g., if the model predicts that an uncontrolled car will continue straight, but it actually turns left, the controlled effects will still be correct if the predictions are consistent for each action. Although it helps, this does not fully solve the issue.
>
> > e_c^goals come from a vae which means they are learned vectors. How on earth do they match exactly the e_c^step that's used to match in algorithms 1 and 2. This seems very fishy and quite a limitation of the method.
>
> We compute the euclidean distance between the two effects and use a threshold to decide if the goal has been achieved. The code refers to this as "has_achieved"; clearly, this function has not been defined and will be added to the paper for better understanding.

---

### Decision · Program_Chairs · 2022-01-12

**Decision:**

Accept (Poster)

**Comment:**

The paper proposes a method for using and learning "controlled" effects for hierarchical reinforcement learning. The paper uses "controllable" states as goal. An interesting aspect of the proposed method is that it allows learning a model of the world that considers only controllable aspects of the environment and hence facilitating exploration and efficient generalization. Even though the reviews are mixed, I like the underlying idea. I encourage the authors to take into account the feedback by the reviewers and also include a "limitations" section discussing various limitations of the proposed method (stronger dependence on the compositional states), and other assumptions required by the proposed method.